# Vaccine Misinformation Detection in X using Cooperative Multimodal Framework

## ABSTRACT

Identifying social media posts that spread vaccine misinformation can inform emerging public health risks and aid in designing effective communication interventions. Existing studies, while promising, often rely on single user posts, potentially leading to flawed conclusions. This highlights the necessity to model users' historical posts for a comprehensive understanding of their stance towards vaccines. However, users' historical posts may contain a diverse range of content that adds noise and leads to low performance. To address this gap, in this study, we present VaxMine, a cooperative multi-agent reinforcement learning method that automatically selects relevant textual and visual content from a user's posts, reducing noise. To evaluate the performance of the proposed method, we create and release a new dataset of 2,072 users with historical posts due to the unavailability of publicly available datasets. The experimental results show that our approach outperforms state-of-the-art methods with an F1-Score of 0.94 (an absolute increase of 13%), demonstrating that extracting relevant content from users' historical posts and understanding both modalities are essential to identify anti-vaccine users on social media. We further analyze the robustness and generalizability of VaxMine, showing that extracting relevant textual and visual content from a user's posts improves performance. We conclude with a discussion on the practical implications of our study by explaining how computational methods used in surveillance can benefit from our work, with flow-on effects on the design of health communication interventions to counter vaccine misinformation on social media. We suggest that releasing a robustly annotated dataset will support further advances and benchmarking of methods.

## CCS CONCEPTS

• **Information systems → Multimedia and multimodal retrieval**.

## KEYWORDS

Vaccine Misinformation, Multimodal Posts, Cooperative Learning

**ACM Reference Format:**

Anonymous Author(s). 2023. Vaccine Misinformation Detection in X using Cooperative Multimodal Framework. In *Proceedings of Make sure to enter the correct conference title from your rights confirmation emai (Conference acronym 'XX).* ACM, New York, NY, USA, 10 pages. https://doi.org/XXXXXXXXX

**Unpublished working draft. Not for distribution.**

## 1 INTRODUCTION

Vaccines are an effective means of preventing and controlling infectious diseases. While access to medical care remains a barrier to vaccine coverage, the dissemination of vaccine misinformation can contribute to the risk of outbreaks [20] and impact the capacity to develop vaccines for diseases such as human papillomavirus (HPV) [43] or COVID-19 [21]. Vaccine misinformation may be associated with potentially harmful attitudes like vaccine hesitancy or behaviors like vaccine refusal and erode confidence in vaccination campaigns [25, 39].

Vaccine misinformation, such as misleading information about effectiveness, emotional narratives about safety, and conspiracy theories, can harm global public health and erode public confidence in vaccination campaigns [21]. By designing computational methods to automatically identify vaccine critical users on social media, public health organizations and social media platforms can improve how they target communication and education interventions to where and when they are most needed [35].

Analyzing multimodal content on social media platforms, including X (formerly known as Twitter), can help identify emerging threats and can then be used to develop more effective communication strategies and identify key areas of concern that need to be addressed to improve vaccine uptake [20, 35]. Multimodal posts on social media that include vaccine misinformation appear to have increased in parallel with the introduction of COVID-19 vaccines [12] and are a popular source of vaccine misinformation [39].

Recent research on the representation of vaccines on social media demonstrated that methods that incorporate both textual and visual information are potentially useful compared to those that use only textual data [4, 39]. However, the limited public availability of annotated multimodal datasets limits our ability to investigate new ways of interpreting and detecting vaccine misinformation.

Recently, various attention-based methods have been proposed to detect vaccine misinformation. For example, Wang et al. [39] presented a multimodal framework with semantic and task-level attention to focus on the essential contents of a multimodal post to detect vaccine misinformation on social media. Shang et al. [32] presented a Duo-generative explainable (DGExplain) method for identifying multimodal COVID-19 misinformation that evaluates the cross-modal relationship between visual and textual content in multimodal content. In another study by Gao et al. [14] presented multi-agent SelectNet (MASN), a reinforcement learning-based method, leveraging pretrained BERT and ResNet models to extract text and image features. MASN utilizes opinion-word and image-region selectors to enhance personality classification collaboratively, outperforming strong unimodal and multimodal baselines. However, using previous methods focused on encoding users' historical multimodal posts adds noise and leads to low performance. This is because a user's historical posts contain a variety of multimodal content, i.e., vaccine relevant and irrelevant posts (Figure 1).

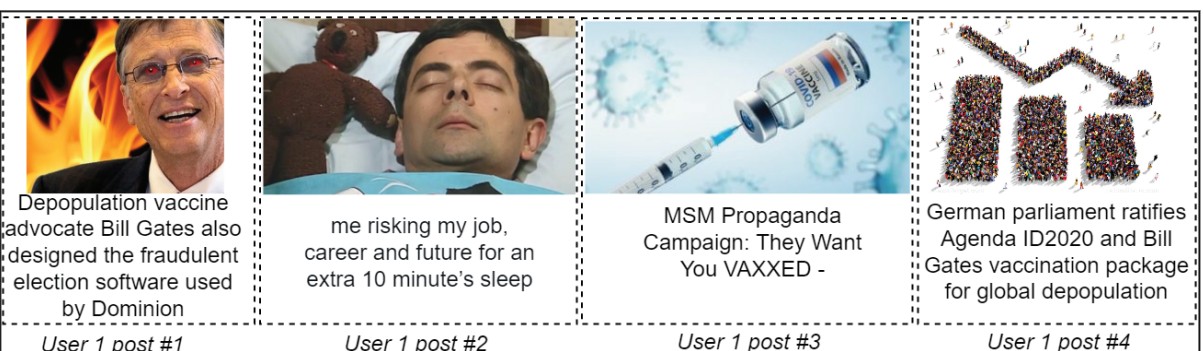

**Figure 1: Examples of the relevant (post #1, post #3, and post #4) and irrelevant (post #2) posts from historical postings of a user who posts vaccine misinformation. User posts are chronologically ordered from left to right.**

In this study, we aim to overcome the above limitations by presenting MM-Vax, a new multimodal data of 2,072 X users and present VaxMine, which uses multi-agent reinforcement learning (MARL) method, which uses two policy gradient agents to simultaneously select texts and images, and evaluate the utility of joint actions based on the classification performance. To address the challenge of determining each selector's contribution in cooperative settings, where joint selections result in global rewards, VaxMine uses a novel cooperative inverse operation multi-agent policy gradients that takes an actor-critic method with differentiated advantages, in which each actor (i.e., text selector or image selector) is trained by following its unique gradient estimated by a critic. We use a two-fold approach. First, we adopt centralized training with decentralized execution, where the critic plays a role only during the learning phase while the actor handles execution. Secondly, we introduce "inverse operation" to benefit each agent. This involves providing each agent with a shaped reward that measures the difference between their current global reward and the reward they receive when taking an opposite action (inverse operation). Our contributions are as follows[1]:

- We construct and release a new multimodal dataset of 2,072 users to detect vaccine misinformation on X.
- We present VaxMine, a cooperative multi-agent reinforcement learning (MARL) based approach where text and image selectors cooperatively extract the vaccine relevant content to identify vaccine misinformation.
- We show that the performance increases from selecting vaccine relevant content from a user's historical posts and VaxMine outperforms SOTA baselines with an F1-Score of 94% and also establishes the generalizability of VaxMine.

## 2 RELATED WORK

### 2.1 Existing Datasets

We know of two multimodal datasets that have been developed for identifying vaccine misinformation using social media. Wang et al. [39] compiled 31,282 posts from Instagram, comprising a single user post containing both textual and visual content and collected between January 2016 and October 2019. Multimodal COVID-19 Vaccine Focused Data Repository (MMCoVaR) [4] is a COVID-19 vaccine dataset that includes 24,184 X posts containing both textual and visual content and collected from February 2020 to March 2021.

---

[1]Code and dataset are available at **Anonymized**

Both of mentioned datasets are not publicly available, whereas we make our dataset publicly available for further research.

### 2.2 Existing Methods

**Prior methods using textual data**: Most research to detect vaccine misinformation has mainly focused on using a textual post and used traditional machine learning methods that include support vector machines (SVM) [2], Naive Bayes ensembles and maximum entropy classifiers [30], and hierarchical SVMs [11]. Recent studies [28, 44] that classify vaccine related posts on social media used bidirectional encoder representations from Transformers (BERT) [8] and its variants to encode a user post.

Other text-based studies [31, 45] that classify a user use different methods to extract textual features and process them sequentially. For instance, Zogan et al. [45] presented DepressionNet, which uses a hybrid extractive and abstractive summarization method to summarize historical posts of a user and then processes them sequentially to identify depression on X. Sawhney et al. [31] used a longformer to extract textual features of all user's posts and sequentially process the historical user posts using Bidirectional Long Short Term Memory (BiLSTM) to identify suicide risk on social media.

**Prior methods using multimodal data**: Previously research has examined the use of a multimodal post for the detection of various tasks such as fake news [10, 19, 37, 38], Sentiment and Emotion [34], COVID misinformation [32], suicide [3], and depression [5, 42]. Limited research has explored multimodal content regarding social media's role in identifying vaccine critical users. Wang et al. [39] presented a multimodal deep neural network with semantic and task-level attention referred to as Seta-Attn for detecting vaccine misinformation on social media. Experiments conducted using unimodal and multimodal in previous studies showed that understanding both modalities is essential for understanding users' opinions accurately.

Recently, researchers also focused on using historical multimodal posts of a social media user for the identification of suicide [3] and depression [5] tasks using different methods to extract textual features of all historical posts of a user and processing them sequentially. For instance, Cao et al. [3] presented SDM where they used Fast-Text embeddings of a user's posts and processed them sequentially using an LSTM to capture the temporal dependencies in the user's history for suicide risk identification on social media. Cheng and Chen [5] presented MTAN, using BERT and inceptionResNet, to obtain historical textual and visual content features and process them

**Table 1: Annotation instructions with top 3 most salient words and the SAGE coefficient. Higher SAGE coefficients indicate salience in the data of that label.**

| Label | Instructions | Most salient words | SAGE coefficient |
|---|---|---|---|
| Misinformation | User posts (text or image or both) contains vaccine misinformation, criticizes vaccines, vaccine conspiracy theories, and cases or statistical conclusions against vaccines. | depopulation
novaccine
vaccinedeath | 1.71
1.70
1.69 |
| Otherwise | User either posts in favor of the vaccine or reports the events or others' opinions objectively related to the vaccine. No content (text or image) against the vaccine. | amid
coronavirus
outbreak | 0.32
0.31
0.30 |

sequentially for depression detection. Gao et al. [14] introduced the multi-agent SelectNet (MASN), a reinforcement learning-based approach that employs pretrained BERT and ResNet models to extract text and image features. MASN employs opinion-word and image-region selectors to collaboratively improve personality classification, surpassing both unimodal and multimodal baseline methods.

The above-mentioned studies explored various encoding, attention strategies, and reinforcement learning techniques. However, these methods alone are insufficient for selecting relevant content, as they may add noise and lead to low performance when applied to the variety of content posted by a user. This is because a user's historical posts contain a variety of multimodal content, including vaccine-relevant and irrelevant posts. We hypothesize that accurately selecting vaccine-relevant content from the variety of posts available can enhance the performance of computational methods in detecting vaccine misinformation on social media.

## 3   DATA

**Data selection and collection:** We augmented a publicly available dataset of X users, released by Muric et al. [27]. Using the post ids, we collected posts that contained both text and images using the X API and grouped the hydrated historical posts by the user.

**Filtration:** We start the filtration step, which includes excluding users with a posting history of fewer than 2 posts or who post in a language other than English. We adopted Optical Character Recognition (OCR) to extract the textual content embedded in the images. Annotators were presented with the original image, the OCR text, and historical posts from a user to annotate the data as per the given annotation instructions.

**Annotation:** We had 8 members on our annotation team, including a combination of both men and women, fluent in English, have degrees ranging from MSc to PhD, as well as expert researchers in natural language processing and computer vision. The annotation required an understanding of both textual and visual content posted by a user.

Each user was annotated independently during the annotation. We instructed annotators to label a user with one of two labels (vaccine misinformation or otherwise). Where annotators disagreed, a new annotator was assigned, and the label was assigned by majority vote. Annotators were provided with the posts in batches to ensure consistency. In a random sample of 600 users, a coefficient of agreement between annotators, i.e., Fleiss' kappa ($\kappa$) was high ($\kappa = 0.86$).

**Instructions:** Annotators were given instructions (Table 1) designed following a previous similar study conducted by Wang et al. [39]. Before starting the annotation process, annotators were requested

**Table 2: Dataset Statistics**

| | |
|---|---|
| Total No. of users | 2,072 |
| Total No. of posts | 30,385 |
| Avg. No. of posts per user | 15 |
| Max. No. of posts per user | 542 |
| Avg. length of posts | 14 |
| Max. length of posts | 124 |
| Class Distribution (%) | |
| Misinformation | 28% |
| Otherwise | 72% |

to read the instructions. To determine whether the annotators understood the annotation instructions, more discussions were held. In particular, annotators were instructed to select one of two labels, i.e., "misinformation" or "otherwise". A user's post that criticizes vaccines, includes vaccine misinformation and conspiracy theories and presents cases or statistics against vaccines is labeled as "misinformation." In contrast, a user who reports the events or talks about vaccines objectively and contains no vaccine related misinformation posts in the entire user post history is labeled as "otherwise." In both cases, a user may contain irrelevant posts (Figure 1). If annotators were unsure about a user, they were instructed to remove those users.

**Data analysis:** Using Sparse Additive Generative (SAGE) [13], we analyze the linguistic variation within labels (Table 1). SAGE implies that by identifying the differentiating words, we can determine the relative relevance of a class to all other classes by comparing word distributions between a target corpus and a reference corpus that use the metric of a log-odds ratio. Because of SAGE's additive nature, we can identify which words significantly impact each label. The word cluster for the "misinformation" label consists of strongly negative words such as 'depopulation', 'novaccine', 'vaccinedeath' as anticipated. As for the "otherwise" label, we notice clear examples of neutral word usage, such as 'amid', 'coronavirus', and 'outbreak'. There is a shift in users' language across both labels, as shown by the most salient words in each class.

**Data statistics:** Based on the above steps, we constructed a new multimodal dataset containing 2,072 X users (28% misinformation and 72% otherwise) with 30,385 total number of posts[2]. A user's average number of posts is 5, with the maximum number of posts

---

[2] We collected posts from over 10,000 X users and randomly shuffled them to ensure that our data encompassed all dates and events. We filtered out users based on specific criteria mentioned earlier and annotated the posts of 2,072 X users following the annotation instructions, which is 20.72% of the total X users collected.

being 542. Similarly, each post has an average of 14 tokens with a maximum of 124 tokens (Table 2).

## 4 METHODOLOGY

**Problem Definition**: Given a collection of posts made by the $u$-th user, represented by $P_u$, containing $T$ pairs of text and images made by a X user. We aim to identify a vaccine misinformation in X.

**Overview of proposed architecture:** Figure 2 illustrates the overall architecture of VaxMine. We present inverse operation-based cooperative multi-agent policy gradients that employ a centralized training framework with decentralized execution by applying a centralized critic and differentiated advantages. Policy gradient agents are used in text and image selectors to determine if a feature should be selected based on the inputs. The critic's gradient estimates are used to train the selectors. Differentiated advantages are represented by rewards that contrast the current global reward with the rewards obtained when each agent's action is substituted by an opposite action (i.e., inverse operation). The domain-specific language model, i.e., COVID Twitter BERT (CT-BERT) [26] and vision transformer (ViT) [9] are used to obtain the text and image features, respectively. The classifier classifies a user based on the features selected by agents.

### 4.1 Features Extraction

*4.1.1 Textual Feature Extraction:* Each text in a post is composed of a sequence of words $w_{t_1}, w_{t_2}, \cdots, w_{t_n}$, where $w_{t_i} \in R^d$ is the $d$-dimensional vector representing the $i$-th word in the $t$-th text, and $n$ is the text's length. We used CT-BERT to compute the continuous representation of posts. CT-BERT have the same architecture as BERT but is trained on COVID-related posts.

*4.1.2 Image Feature Extraction:* ViT is used to extract image features in our method. ViT applies a transformer architecture to image patches. Position embeddings are incorporated for each fixed-size patch, and the resulting vector sequence is fed to a classic transformer encoder.

Above generated textual and image features are then fed to the inverse operation-based cooperative MARL module.

### 4.2 Inverse Operation Based Cooperative MARL

In the following discussion, we introduce two agents (i.e., text selector and image selector) that we use to select a vaccine relevant text and image from users' historical multimodal posts.

**Text Selector and Image Selector:** In one event, one user's historical posts $P_u$ belong to the user's sequential posts. We used the features extracted $h_t^{text}$ and $h_t^{image}$ for the text and image selectors in step $t \in T$. We use $a_t \in A = \{0, 1\}$ to detect whether or not to select the feature at step $t \in T$. During implementation, the local action observation histories must be used to learn the policy $\pi(a : T)$. To understand the full history of the selectors, we used Bidirectional Gated Recurrent Unit (BiGRU) [7] to model them. As a result, the agents' policy $\pi(a : T)$ is formulated as:

$$\pi^e(a_{1:T}) = \prod_{t=1}^{T} \pi^e(a_t | s_t) \tag{1}$$

$$g_t^e = BiGRU(h_t^e, g_{t-1}^e)$$

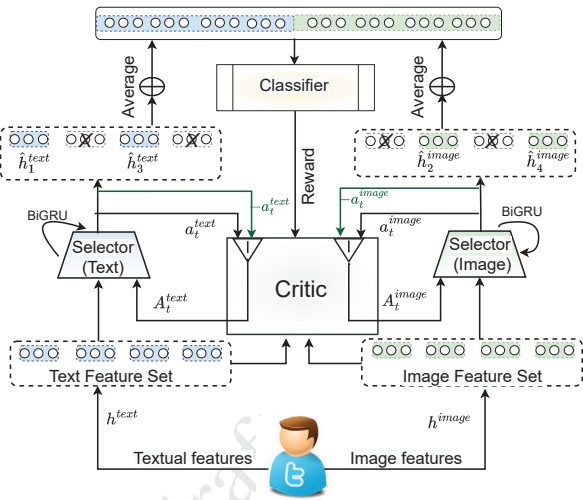

**Figure 2: Overview of our proposed architecture. To identify selector $e$'s advantage $A_t^e$, we compare the current global reward with the reward obtained when an agent's action is changed with an inverse action $-a_t^e$ at each $t$-step.**

$$\pi^e(a_t | s_t) = (1 - a_t^e) * (1 - \sigma(MLP(g_t^e))) + a_t^e * \sigma(MLP(g_t^e)),$$

where $e \in \{text, image\}$. The BiGRU's hidden state is denoted by $g_t$, and MLP denotes a multilayer perceptron layer. $\sigma(\cdot)$ is a sigmoid function that turns $g_t$ into a probability. Following that, the selector samples an action to determine whether to select the feature ($a_t = 1$) or not ($a_t = 0$). The feature $h_t^e$ will be modified as $\hat{h}_t^e$ and added to $H_{indi}^e$ if a particular feature is selected. $H_{indi}^e$, a subset of the user representation, is then used to make predictions.

**Classifier:** For binary classification, a subset of selected features is used at each step $t$ in one event. We combined $H_{indi}^{text}$ and $H_{indi}^{image}$ using average operation to generate a user representation. Thus, we used two fully connected layers (MLPs) and a dropout operation to process users' representation. Following the output of the final layer, a non-linear sigmoid layer is used to predict the probability distribution.

where $\oplus$ indicates the concatenation operation and $\hat{y}_u$ denotes the probability distribution of the prediction. We can reward the selector based on the likelihood of the ground truth, i.e., $P_\gamma(y = \hat{y}_u)$ to take better actions. The actor-critic technique can be used to apply the change in the $P_\gamma(y = \hat{y}_u)$ after upgrading its sets with the newly selected instances as the unified temporal difference error [41].

$$r_t = P_\gamma(y = \hat{y}_u | o_{t+1}) - P_\gamma(y = \hat{y}_u | o_t)$$

$$L_t(\theta_c) = [\gamma_t + (H_{indi}^{t+1}, \Pi_{t+1}, A_{t+1}) - Q(H_{indi}^t, \Pi_t, A_t]^2 \tag{2}$$

where $H_{indi}^t = H_{indi}^{text} \oplus H_{indi}^{image}$, $\Pi = \pi^{text} \oplus \pi^{image}$, and $A = a^{text} \oplus a^{image}$. Whereas using similar advantages makes it hard to conclude the contribution of each selector. As a result, differentiating the advantages is essential.

**Differentiated Advantages via Inverse Operation:** In our proposed settings, different rewards can be implemented using a centralized critic. Although our method learns a centralized critic that estimates Q-values for joint action based on the central state $H_{indi}^t$, we can

**Algorithm 1** Inverse Operation Based Cooperative MARL

1: Initialize the critic network randomly $Q(S, \pi, \mathbf{a}|\theta_Q)$ and 2 selectors $\pi(s|\theta_\pi^e)$ with weights $\theta_Q$ and $\theta_\pi^e$.

2: Initialize target network $Q'$ and $\pi'$ with weights $\theta_{Q'} \leftarrow \theta_Q$, $\theta_{\pi'}^e \leftarrow \theta_\pi^e$. Replay buffer initialization $R$

3: **for** event = 1, M **do**

4:     Obtain the initial observation state $h_1^e$

5:     **for** $t = 1, T$ **do**

6:         Select action $a_t^e = \pi(h_t^e|\theta_\pi^e)$ as per the current policy

7:         Perform action $a_t^e$ and observe the likelihood of ground truth $\Pr(y = \hat{y}_u|o_t)$ and observe the new state $h_{t+1}^e$

8:         Perform action $a_{t+1}^e$ and observe the likelihood of ground truth $\Pr(y = \hat{y}_u|o_{t+1})$, as a result, obtain the reward $r_t = \Pr(y = \hat{y}_u|o_{t+1}) - \Pr(y = \hat{y}_u|o_t)$

9:         Store transition $(H_{indi}^t, A_t, r_t, H_{indi}^{t+1})$ in $R$

10:        Sample N random transitions from a minibatch $(H_{indi}^i, A_i, r_i, H_{indi}^{i+1})$ from $R$

11:        Set $z_i = r_i + \gamma Q'(H_{indi}^{i+1}, \Pi_{i+1}, A_{i+1})$

12:        Minimize the loss to update the critic:

$$\mathcal{L}(\theta_Q) = \frac{1}{N} \sum_i \left[ z_i - Q(H_{indi}^i, \Pi_i, A_i|\theta_Q) \right]^2$$

13:        Updating selectors with differentiated advantages:

$$A^e(H, \Pi, A) = Q(H, \Pi, A) - (H, \Pi, (-a^e, a^{-e}))$$

$$\nabla_{\theta_\pi^e} J(\theta_\pi^e) = \nabla_{\theta_\pi^e} \log \pi(a_t^e|h_t^e) A^e(H, \Pi, A)$$

14:        Updating target networks:

$$\theta_{Q'} = \tau\theta_Q + (1-\tau)\theta_{Q'}, \theta_{\pi'}^e = \tau\theta_\pi^e + (1-\tau)\theta_{\pi'}^e$$

15:     **end for**

16:     Minimize cross-entropy loss:

$$J(\theta_C) = -[y_u \log \hat{y}_u + (1 - y_u) \log(1 - \hat{y}_u)]$$

17: **end for**

give each agent a distinct advantage by contrasting the global reward to the reward when the agent takes the opposite action. Intuitively, when the Q-value of the gold action is subtracted from the Q-value of the opposite action, this method provides a positive reward. Formally, we can then calculate an advantage function for each selector $e$ by comparing the current action's Q-value $a^e$ to an inverse operation baseline that takes the opposite action $a^e$ while maintaining the other agent's action $a^{-e}$ constant:

$$A^e(H, \pi, A) = Q(H, \Pi, (A^e, a^{-e})) - Q(H, \Pi, (-a^e, a^{-e})) \quad (3)$$

As a result, $A^e(H, \Pi, A)$ calculates a baseline and a centralized critic for each agent and each advantage. Thus, Algorithm 1 can be used to optimize the model further.

## 5 EXPERIMENTS

### 5.1 Baselines

We compared our method with state-of-the-art (SOTA) unimodal (text only and image only) and multimodal methods including recent LMMs. Our comparative methods include those used to identify vaccine misinformation and other multimodal classification tasks.

#### 5.1.1 *Unimodal Models*.

- **Text only**: We adopted two different methods to encode textual features. (i) We concatenated all historical posts and used LSTM [17], GRU [6], a BERT [8] and GPT-4 [29] to obtain textual features. (ii) We also used DepressionNet [45], a text-only method designed to identify a user's depression using historical posts.
- **Image only**: We used DenseNet [18], VGGNet [33], ResNet [16] and vision transformer (ViT) [9].

#### 5.1.2 *Multimodal Models:*

- **Mid fusion methods**: For mid-fusion-based multimodal methods, we concatenated all textual and visual contents from users' historical posts and trained separate models on the textual and the visual, BERT and ResNet (ResNet+BERT), respectively, and then we combined them by taking the output of the second-to-last layer of ResNet for the visual part and the output of the [CLS] token from BERT, and we fed them into an MLP. We also used ResNet and fasttext (ResNet+fasttext) to extract the visual and textual features from users' historical posts and concatenated features for classification.
- **VisualBERT** [22]: We concatenated all textual and visual contents from users' historical posts and used a VisualBERT that is trained using a multimodal objective and tested on a wide range of multimodal tasks.
- **Bimodal variational autoencoder (MVAE)** [19]: Encoded representation of multimodal news data is fed to a MVAE to detect fake news.
- **Event adversarial neural network (EANN)** [37]: Uses textual and visual features to train an event-based discriminator to identify fake news using social media multimodal data.
- **Suicide Detection Model (SDM)** [3]: FastText embeddings are employed to encode historical user posts, fed into an LSTM layer following an attention layer for suicide identification.
- **User Preference-aware Fake Detection (UPFD)** [10] detects fake news by modeling social context, and historical posts of a social media user.
- **Multimodal time-aware attention networks (MTAN)** [5]: BERT and InceptionResNet are used to obtain features of historical textual and visual content. Extract features of historical visual and textual data are then fed to a time-aware LSTM layer, followed by attention for depression detection.
- **Large Multimodal Models (LMMs)**: For LMMs, we used LLaVA [23], MMGPT [15] and CogVLM [36].
- **Duo-generative explainable (DGExplain)** [32]: a generative method for identifying multimodal COVID-19 misinformation that evaluates the cross-modal relationship between visual and textual content in multimodal content.
- **MASN** [14]: a tailored reinforcement learning approach to integrate the opinion-word and image-region selection strategies to select information for opinion-word and image-region features for multimodal personality classification.
- **Seta-Attn** [39]: We also compared results with Seta-Attn, a recently proposed method for identifying vaccine misinformation. Seta-Attn uses visual and textual content and semantic- and task-level attention to focus on the essential contents of a post that indicate vaccine misinformation multimodal posts.

## 5.2 Experimental settings

We reported an average of 10-fold cross-validation for all the results. We used a similar experimental setup for all baseline methods, followed the original model settings described in their paper, and used grid search optimization to obtain the optimal hyperparameters. We used the base version of pretrained language models. We used an Adam optimizer with a learning rate of 0.001 to optimize our model. We padded posts with different lengths and trained our model for 150 epochs with early stopping with the patience of 10 epochs.

## 6 RESULTS

**Comparison with Baselines:** Table 3 presents the performance of our method when compared to the SOTA methods. As expected, the performance of text-only methods was better than that of image-only methods, given that text contains more explicit vaccine misinformation than visual information in posts. Moreover, transformer-based models such as BERT, CT-BERT and GPT outperform GRU and LSTM, which is expected because these models can better capture contextual representation than LSTM and GRU. DepressionNet performs better when compared to other text-only baselines, including GPT. We attribute this increase to the possibility of better capturing the users' historical posts. In addition, we also note that the performance of both (text-only and image-only) models is low, i.e., no higher than an F1-Score of 73%. Hence, the performance obtained using text-only or image-only methods is less desirable in identifying vaccine misinformation in multimodal data. Following that, we demonstrate how combining texts and images helps improve performance when identifying vaccine misinformation.

We observed higher performance in multimodal methods than text or image-only methods (Table 3). Further, the SOTA attention-based multimodal methods, i.e., VisualBERT and Seta-Attn, performed slightly better than ResNet + BERT and ResNet + fasttext. However, according to our analysis, only a small percentage of users' historical posts contain relevant content to identify vaccine misinformation. As a result, multimodal methods based on attention may have difficulty capturing the relevant content. We adopted an RL-based selection strategy to address the above challenge. The proposed method achieves an F1-Score of 0.94, an absolute increase of 13% when compared to Seta-Attn (best baseline), designed to capture vaccine misinformation using multimodal data. Our results validate that differentiated advantages outperform attention-based (Seta-Attn) multimodal methods. We attribute the performance gains to selecting relevant content from users' historical posts, and the use of differentiated content also improves performance compared to attention-based methods for identifying vaccine misinformation. We also observe that the performance of other multimodal methods, including LMMs (i.e., MVAE, EANN, SDM, UPFD, DGExplan, MTAN, CogVLM, LLaVa and MMGPT) are less desirable in detecting vaccine misinformation on multimodal social media posts. The reason is that these methods encode both relevant and irrelevant content from the users' historical multimodal posts on social media that add noise and result in low performance.

**Post-wise comparison**: Figure 3 shows the results of our method with varying numbers of posts compared to the best baseline method (Seta-Attn). We showed that the F1-Score of our method increases monotonically and outperforms Seta-Attn in each number of posts

**Table 3: Performance comparison. * shows that our method obtained a significant ($p < 0.05$) improvement over best baseline (underlined) under Mann–Whitney U test.**

| Modality | Method | F1-Score | Precision | Recall |
|---|---|---|---|---|
| Text | GRU | 0.69 | 0.68 | 0.70 |
| | LSTM | 0.68 | 0.69 | 0.66 |
| | BERT | 0.71 | 0.70 | 0.71 |
| | CT-BERT | 0.72 | 0.71 | 0.71 |
| | GPT-4 | 0.72 | 0.70 | 0.72 |
| | DepressionNet | 0.73 | 0.72 | 0.73 |
| Image | DenseNet | 0.65 | 0.71 | 0.64 |
| | ResNet | 0.66 | 0.68 | 0.66 |
| | VGGNet | 0.64 | 0.72 | 0.65 |
| | ViT | 0.67 | 0.74 | 0.67 |
| Multimodal | ResNet+BERT | 0.75 | 0.75 | 0.76 |
| | ResNet+fasttext | 0.74 | 0.78 | 0.71 |
| | VisualBERT | 0.77 | 0.76 | 0.72 |
| | MVAE | 0.78 | 0.77 | 0.78 |
| | EANN | 0.79 | 0.78 | 0.79 |
| | SDM | 0.77 | 0.77 | 0.77 |
| | MTAN | 0.79 | 0.78 | 0.79 |
| | CogVLM | 0.78 | 0.78 | 0.78 |
| | LLaVA | 0.80 | 0.78 | 0.79 |
| | MMGPT | 0.77 | 0.77 | 0.78 |
| | UPFD | 0.79 | 0.79 | 0.79 |
| | DGExplain | 0.80 | 0.79 | 0.80 |
| | MASN | 0.78 | 0.80 | 0.79 |
| | Seta-Attn | 0.81 | 0.81 | 0.81 |
| | Proposed | 0.94* | 0.94* | 0.94* |

evaluated. We also demonstrated that our method achieved the highest F1-Score within the first 5 posts, which is the average number of historical posts for a user in our dataset. We observed that the F1-Score of Seta-Attn increased from 62% to 80% when evaluated on the first 20 historical user posts and achieved a maximum F1-Score of 81% when evaluated on all historical posts. In contrast, the performance of our method remains stable. Further, we observed that increasing the number of posts in a user's history does not greatly enhance model performance. We attribute the improvement in saturation to a small number of users with more than 20 posts.

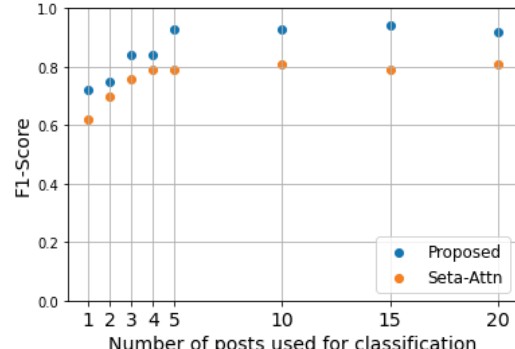

**Figure 3: Post-wise analysis: Proposed v/s Seta-Attn**

**Table 4: Ablation analysis: Proposed w/o selector shows the result of using all posts, i.e., without selector module. Proposed w/o image features and proposed w/o text features indicate the results without image and test features in our method. *shows that our method obtained a significant ($p < 0.05$) improvement over other variants of our method under Mann–Whitney U test.**

| Method | F1-Score | Precision | Recall |
|---|---|---|---|
| Proposed | 0.94* | 0.94* | 0.94* |
| Proposed w/o selector | 0.86 | 0.86 | 0.86 |
| Proposed w/o image features | 0.84 | 0.84 | 0.85 |
| Proposed w/o text features | 0.76 | 0.81 | 0.78 |

## 6.1 Analysis

**Ablation analysis:** An ablation analysis (Table 4) shows the effectiveness of each of the new components we included in our method contributed to the overall performance. The F1-Score drops (from 0.94 to 0.86) when we remove the selectors and feed both relevant and irrelevant multimodal content, indicating the importance of selecting texts and images that are more useful to classify vaccine critical users. This drop in F1-Score validates our motivation to select relevant content, which is ignored in previous studies. Further, we first removed image feature extraction from our method to demonstrate the importance of multimodal features. The performance dropped to 0.76 when we removed the text feature module from our proposed method. These results show that textual information plays a more important role than visual information in identifying vaccine critical users. Similarly, the F1-Score drops to 0.84 when removing the image feature extraction module from our proposed method. This drop in performance in both (removal of text and image) showed the importance of using both text and image content and validates our motivation that both visual and textual content should be jointly considered to make accurate inferences. Hence, we conclude that the strengths of the proposed method lie in the combination of multimodal features and selective posts from a user's historical posts that contribute to increased performance.

**Robustness analysis in realistic settings:** Considering realistic settings where there could be a small percentage of vaccine critical users, we conducted a robustness analysis of the proposed method on different percentages of vaccine critical users (Figure 4). To conduct the robustness analysis, we varied the percentage of users from 10% to 90% at increments of 10% and observed that our method achieved better performance despite of low percentage of vaccine critical users. We also note that when the percentages do not reach 50%, the best baselines method (Seta-Attn) performs poorly. Our method outperformed Seta-Attn with an F1-Score of 77% (an absolute increase of 16%) when tested on 10% of the data. Hence, we can conclude that our method is robust in realistic scenarios.

**Generalizability test:** A generalizability test on other datasets used by Cao et al. [3] for suicide detection and Cheng and Chen [5] for depression detection tasks that use multimodal data with the user's historical post shows that our method outperformed the best results reported in [3] for suicide detection and in [5] for depression detection and Seta-Attn by an absolute increase 4% and 2% on both tasks (Figure 5). Our generalizability test concludes that our

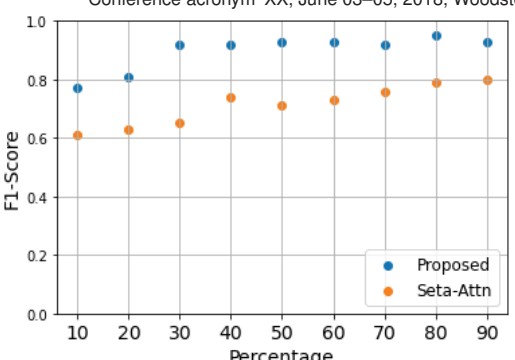

**Figure 4: Proposed v/s Seta-Attn trained on data with varying percentages of vaccine critical users to validate the robustness.**

method is generalizable and outperforms the state-of-the-art and best baseline (Seta-Attn) in suicide and depression classification tasks.

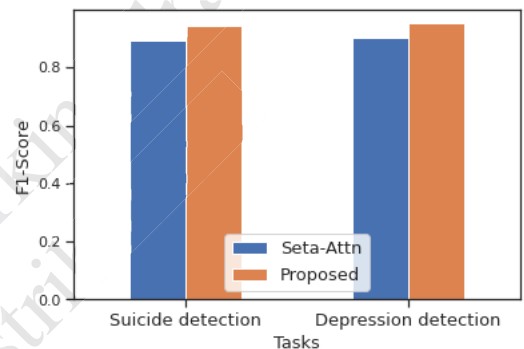

**Figure 5: Generalizability: proposed v/s the baseline method on Suicide detection and Depression detection task.**

**Multimodal baselines with selector:** We also investigated the effectiveness of our selector module in multimodal baselines (Figure 6). We observed that the performance of each tested baseline increased (ranging from 2% to 8%). Seta-Attn obtained the highest F1-Score of 0.86 after incorporating a selector module that selects the relevant content from the user's historical posts. This improvement in baseline results confirms the usefulness of using a selector that selects relevant content from the user's historical posts.

**Qualitative analysis:** Figure 7 shows an example correctly predicted by our method because it selects vaccine relevant posts from users'

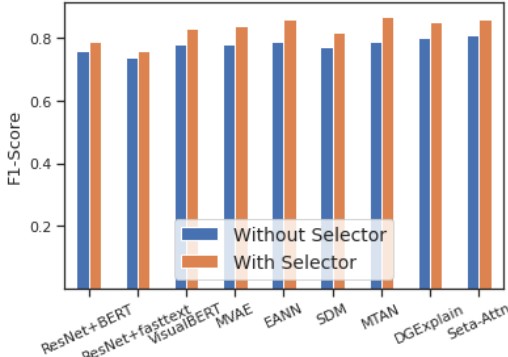

**Figure 6: Baselines (multimodal) with and without selector**

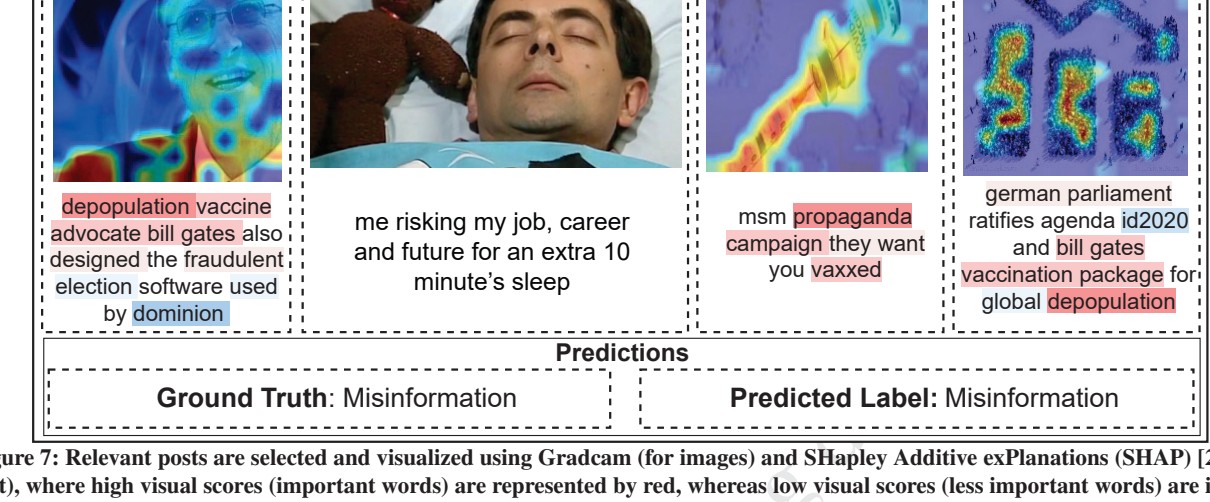

**Figure 7: Relevant posts are selected and visualized using Gradcam (for images) and SHapley Additive exPlanations (SHAP) [24] (for text), where high visual scores (important words) are represented by red, whereas low visual scores (less important words) are in blue. The bottom row shows the prediction results using our method.**

historical multimodal posts and focuses on the important features. For example, for images, our method captures the face (of Bill Gates), a syringe, and clusters of people (representing the decrease in population). For text, words highlighted in red such as depopulation, vaccine advocate, Bill Gates, propaganda, etc., contribute more to the final prediction. We also show that our selector module selects relevant content and ignores the irrelevant content (user post #2) from the user's historical posts. Further, from the prediction analysis, we observed that our method correctly classified a user by selecting relevant posts and leveraging visual and textual data.

**Error analysis:** When we examined specific examples of incorrect predictions, we found they mostly corresponded to one of three types. These were posts where there was insufficient information, i.e., both image and text do not contain words or user opinions against vaccines, the OCR failed to detect the words, and posts that require external knowledge.

## 7 PRACTICAL IMPLICATION

Social media posts that include text and images can spread quickly and may contribute to normalizing vaccine misinformation, as well as making them seem more common than they are. Tools for quickly determining which users are posting vaccine misinformation at scale are important for catching the spread of vaccine critical content, including misinformation and other efforts at undermining public health actions. Tools that make use of methods for classifying users and posts from multimodal data can then support public health organizations through early signaling and identification of emerging misinformation threats, which in turn can guide the design and deployment of countermeasures such as communication interventions delivered via social media.

Our results show that we can more accurately predict which users are engaging with and spreading vaccine misinformation by looking at their historical posts. This can support actions taken by social media platforms, including more precise flagging of posts so that

other social media users are warned about the content in advance or reducing the visibility of the content by downranking or not recommending it in user timelines. We expect the work will also have an impact beyond vaccination and may be useful for other scenarios where multimodal data are used to spread misinformation.

## 8 CONCLUSION

We study the problem of detecting vaccine misinformation in X. Our contribution is to release a new multimodal dataset (MM-Vax) with historical posts of 2,072 X users and a novel reinforcement learning-based method (VaxMine) that selects the relevant content from the historical posts of a user for better classification performance. Our experimental results showed that our method outperformed SOTA methods. We further showed the generalizability of our method on other multimodal tasks. We demonstrated that understanding both modalities and selecting relevant content from users' historical posts is important to identify vaccine misinformation on social media.

## ETHICAL CONSIDERATIONS

We carefully considered potential ethical issues in this work: (i) protecting users' privacy and (ii) avoiding potentially harmful uses of the proposed dataset. The X privacy policy explicitly authorizes third parties to copy user content through the X API. We follow the widely accepted social media research ethics policies that allow researchers to utilize user data without explicit consent if anonymity is protected [1, 40]. Any metadata that could be used to specify the author was not collected. Due to the subjective nature of annotation, we expect some biases in the distribution of labels and our gold-labelled data. Hence, any biases that may be found there are unintentional. In addition, all content is manually scanned to remove personally identifiable information . The annotated data we release include de-identified publicly available posts from X, where users understand public access and there is no expectation of privacy. Hence, no ethical approval is required.

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
