# OpenReview forum: "Vaccine Misinformation Detection in X using Cooperative Multimodal Framework"
_acmmm.org/ACMMM/2024/Conference — MM2024 Oral_

### Official Review · Reviewer_2DSd · 2024-05-05

**Rating:** 4
**Confidence:** 3

**Summary:**

This paper releases a new multimodal dataset on vaccine misinformation on X and presents VaxMine, a multi-agent reinforcement learning method, to identify vaccine misinformation. The author also evaluates their approach’s robustness and generalizability.

**Strengths:**

1. This paper focuses on a meaningful research topic - vaccine misinformation detection.
2. The data collection process follows a rigorous process and is presented with details.
3. The author presents a novel approach to dynamically select relevant modality using reinforcement learning.

**Limitations:**

1. In l.355, it is not clear what the vaccine misinformation is in the problem definition.
2. In l.445, is there an equation missing? Also, some equations are center aligned, while some are left aligned. Ideally, they should be aligned in the same fashion. Moreover, a minor thing is that, there shouldn’t be an indent before some of the “where”, because the sentence is not finished yet.
3. In Table 3, the multimodal baselines are all having similar performances (around 0.8). Is it possible to offer some insights for this observation? Also, it seems that BERT + ResNet are used for multimodal baseline models, while CT-BERT + ViT are used for the proposed approach. Is this a contributing factor to the model’s performance? Minor: it would be better if more precision places will be used in the tables, e.g. 3 post-decimal digits.
4. In Figure 7, it’s not so clear how it is determined which post is relevant (selected). Because the proposed approach is not designed for selecting post, instead it selects relevant visual/textual modality. Also, it seems that the outputs of your approach is the sequence of actions (0 or 1) and the classification result. What was the input to Gradcam and SHAP to obtain the region-level and word-level attention of the proposed model?

**Suitability:**

3

---

### Official Review · Reviewer_tKXr · 2024-05-19

**Rating:** 5
**Confidence:** 4

**Summary:**

The authors propose VaxMine, a cooperative multi-agent reinforcement learning (MARL) based approach where text and image selectors cooperatively extract the vaccine relevant content to identify vaccine misinformation. The authors also construct their own dataset, MMVax. They conduct comparative experiments with text-based, image-based, and 13 multimodal-based baselines on the MMVax.

**Strengths:**

- The paper is generally well-written and easy to follow.

- It is nice that the authors construct their own dataset, MM-Vax.

- The authors conduct comparative experiments with six text-based approaches, four image-based approaches, and 13 multimodal-based approaches. The improvements obtained by the authors' approach are supported by statistical significance.

- The authors give not only successful examples and but also error analysis.

- It is nice that the authors give practical implication and ethical considerations.

**Limitations:**

- The authors need to distinguish vector, matrix from scalars.
The equations in Sec 4.2 are little bit unclear. If the authors use the same font for vector, matrix, and scalars, they should use bold fonts for the vectors and matrices to distinguish them from scalars.


- The authors need to select baselines carefully.
It is very nice that the authors conduct comparative experiments with six text-based approaches, four image-based approaches, and 13 multimodal-based approaches. But especially in multimodal models, the newer approaches may have already outperform older ones. In addition, [10] is a short paper, indicating that it is potentially a weak baseline. So the reviewer would like to suggest the following important work on fake news detection as social-context baseline:

Van-Hoang Nguyen, Kazunari Sugiyama, Preslav Nakov, Min-Yen Kan:
"FANG: Leveraging Social Context for Fake News Detection Using Graph Representation" (CIKM 2020, Best Paper Award)


- The authors need to enrich references.
The authors need to cite the following important works on BiLSTM, LSTM, BERT in Sec 2.2 and Adam Optimizer in Sec 5.2:

[BiLSTM]
Alex Graves and Jurgen Schmidhuber:
"Framewise phoneme classification with bidirectional lstm and other neural network architectures"
Neural Networks, 18(5), pages 602–610, 2005.

[LSTM]
S. Hochreiter and J. Schmidhuber:
"Long Short-Term Memory"
Neural Computation, 9(8):1735–1780, 1997.

[BERT]
J. Devlin, M.-W. Chang, K. Lee, and K. Toutanova:
"BERT: Pre-training of Deep Bidirectional Transformers for Language Understanding" (NAACL 2019, Best Long Paper)

[Adam optimizer]
Diederik P Kingma and Jimmy Ba:
"Adam: A Method for Stochastic Optimization"
In Proc. of the 3rd International Conference on Learning Representations (ICLR2015)


The following state-of-the-art would be helpful for improvement in the authors' work:
Maxwell A. Weinzierl and Sanda M. Harabagiu:
"Identifying the Adoption or Rejection of Misinformation Targeting COVID-19 Vaccines in Twitter Discourse" (WWW 2022)


Furthermore, they need to format "References" section carefully.
Page numbers are missing in some references.They also need to explore whether the six arXiv papers have been officially published in a conference or a journal. For example, the reviewer found that [26] and [33] have been published in Frontiers in Artificial Intelligence and ICLR, respectively.

Martin Müller, Marcel Salathé, Per Egil Kummervold:
"COVID-Twitter-BERT: A natural language processing model to analyse COVID-19 content on Twitter"
Frontiers in Artificial Intelligence, Volume 6, 2023.

Karen Simonyan, Andrew Zisserman:
"Very Deep Convolutional Networks for Large-Scale Image Recognition" (ICLR 2015)


- It would be much better to correct the following minor errors in the paper:
Line 445: where \oplus indicates ... => \oplus indicates ...
(It seems that "where" is unnecessary.)

**Suitability:**

3

---

### Official Review · Reviewer_eNis · 2024-05-24

**Rating:** 5
**Confidence:** 3

**Summary:**

This paper proposes VaxMine, a cooperative multi-agent reinforcement learning method designed to identify social media posts spreading vaccine misinformation by analyzing users' historical posts. Furthermore, this paper proposes a new dataset of 2,072 users with historical posts for further researches.

**Strengths:**

1. The motivation is clear: modeling users' historical posts for a comprehensive understanding but may contain noise. To address this gap, this paper proposes a method and a dataset to exploring the area of leveraging users' historical posts for vaccine misinformation detection.
2. Extensive experiments demonstrates the effectiveness of the method VaxMine.
3. The baseline is very sufficient while the experiment is also complete, making the work relatively solid.
4. The authors makes a detailed discussion about ethics in Ethical Considerations.

**Limitations:**

1. The dataset only consists of  2,072 users and 30,385. The limited size of users may effect the generalizability of the findings. Furthermore, it's also limited for future research. Will this dataset be public and keep enlarging its size?
2. While the multi-agent reinforcement learning method shows promise, it iscomplex and computationally intensive. This might limit its practical application.

**Suitability:**

3

---

### Official Review · Reviewer_XkiW · 2024-05-27

**Rating:** 2
**Confidence:** 3

**Summary:**

This paper focuses on the task of Vaccine Misinformation Detection. Concretely, given a user, this user will produce a collection of posts, and this work aims to detect whether this user will produce Vaccine Misinformation or not. This paper claims existing work cannot filter out noise posts. To solve this problem, this work proposes MARL, an RL-based framework to extract the vaccine-relevant posts first and then to integrate these posts for misinformation classification.

**Strengths:**

The proposed method achieves a new SOTA.

This paper collects all historical posts for each post based on a publicly available dataset consisting of posts to construct a new dataset.

**Limitations:**

More experiments are needed to verify the necessity of historic posts. For example, the authors need to compare BERT with the original post proposed in Muric et al. and BERT with historical posts.

More experiments on other datasets or tasks are needed to verify the proposed method's effectiveness. The authors only conduct experiments on one dataset.


The writing of this paper needs further improvements, especially in the method section. For example, in lines 393-394,  the authors do not introduce how to obtain h_t^test in detail, and Eq 2 is wrong. Moreover, the format of the paper is questionable, like lines 456 and 445.


 The topic of Vaccine Misinformation is outdated.

**Suitability:**

3

---

### Official Review · Reviewer_Jd6u · 2024-05-29

**Rating:** 5
**Confidence:** 2

**Summary:**

This paper investigates the problem of vaccine misinformation detection. It starts by collecting a new multimodal dataset containing historical posts from 2,072 users of social media X. It then proposes a new reinforcement learning-based approach to this dataset, which selects relevant content from users' historical posts for better classification performance. Finally, the experimental results show that the proposed method is superior to the SOTA baseline models.

**Strengths:**

S1: Clear presentation and easy to follow;
S2: Dataset is augumented for research.

**Limitations:**

There are some comments:
1. This paper is difficult to read. In addition to the repetition of some phrases, some important information is missing, hindering the reading of the paper. It would also benefit from punctuation application/grammar check and careful proofreading. For example, in the title "using" --> "Using", lines 320 to 324 "misinformation."”-->“"misinformation".” There are many similar problems in the paper. Many complicated long sentences are difficult to read, for example, line 132, line 137...

2. The paper often describes "Each text", "t-th text" and so on, which seems inappropriate, and should be used as sentences or words. There is also something wrong with mathematical equation 2 and the author should revise it carefully. There is also a font inconsistency between the three-level headings, which also exists in "5.1.2 Multimodal Models", and "text only" and "text - only" the use of chaos.3. Papers often say ". The feature ℎ𝑒𝑡 will be modified as ℎˆ𝑒 t …", but it's not clear what kind of action was used to modify it.

3. Line 516 of the paper has "unimodal (text only and image only) …", (text only and image only) is multimodal, should be written "(text only or image only)". Line 728 of the paper has, "This drop in performance in both (removal of text and image) …", The text and the image are all removed, so what's the model input?

4. Authors should revise the non-standard references, e.g. [40].

**Suitability:**

3

---

### Meta-Review · Area_Chair_eM7y · 2024-07-02

**Recommendation:** Accept (Oral)
**Confidence:** 4

**Metareview:**

This submission focuses on detecting text-image vaccine misinformation on the Twitter(X) platform. To effectively select the relevant textual and visual content from users' posts, a multi-agent RL method is proposed. Experiments on one vaccine misinformation dataset (constructed by the authors themselves) and two datasets on other tasks show the effectiveness of the proposed method.

All reviewers believe that the newly constructed dataset is valuable and the performance improvement of the proposed method is clearly a positive contribution. The experiment is also extensive, with many baselines and rich analysis. Though most of the reviews are positive, there are still some issues to be tackled, listed as:

- Writing issues: Reviewers Jd6u and tKXr think that this paper is written well and easy to follow, but Reviewers XkiW and 2DSd found some presentation issues. I confirm the existence of these issues and this paper requires very careful copyediting. The presentation quality of the current version is not acceptable to be published. In L454, a right bracket (")") is missing. In Eq. (1), there is no 𝜋 (𝑎 : 𝑇 ). The authors claimed that h_t^{text} is clearly shown in Figure 2, but I could only find h^{text} and \hat{h}_1^{text}. The omission of the subscription makes the text and the figure consistent enough. The symbol system used in the current version will significantly lower the reading experience of readers and there is no evidence showing that the re-organization is easy to do and will be done.
- Conducting experiments on three datasets of three distinct tasks: The authors claimed that the experiments were conducted on three datasets, which is contrary to the statement from Reviewer XkiW. This is true but also weird, as the title has shown the focus of this paper, i.e., Vaccine Misinformation Detection. I could hardly see suicide and depression as special types of misinformation. The motivation for including three datasets is not supported well.

Overall, I will recommend the acceptance of this submission but with a very heavy hesitation given the manuscript quality.